# MEG-XL: Data-Efficient Brain-to-Text via Long-Context Pre-Training

**Dulhan Jayalath** [1]    **Oiwi Parker Jones** [1]

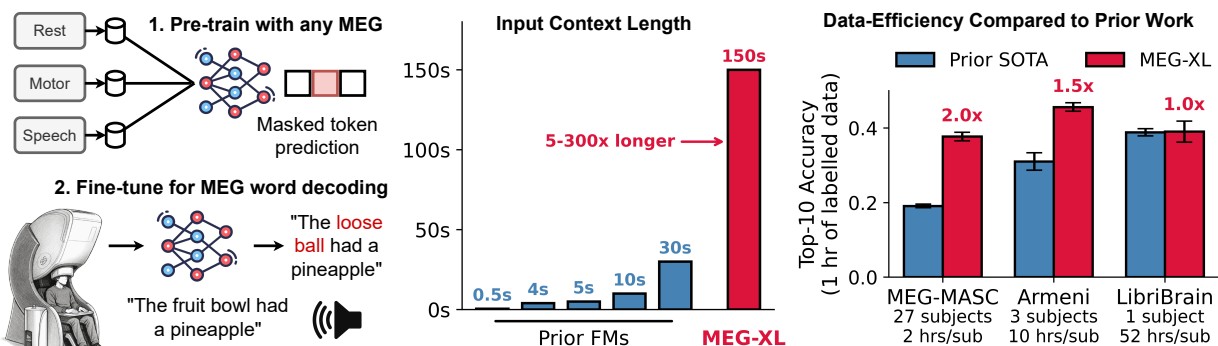

*Figure 1.* MEG-XL introduces long-context MEG pre-training. When fine-tuned, this approach generalises to decoding words in brain-to-text with less labelled subject data than required by the supervised state-of-the-art (SOTA) and brain foundation models (FMs).

## Abstract

Clinical brain-to-text interfaces are designed for paralysed patients who cannot provide extensive training recordings. Pre-training improves data-efficient generalisation by learning statistical priors across subjects, but these priors critically depend on context. While natural speech might unfold gradually over minutes, most methods pre-train with only a few seconds of context. Thus, we propose *MEG-XL*, a model pre-trained with 2.5 minutes of MEG context per sample, 5-300× longer than prior work, and equivalent to 191k tokens, capturing extended neural context. Fine-tuning on the task of word decoding from brain data, MEG-XL matches supervised performance with a fraction of the data (e.g. 1hr vs 50hrs) and outperforms brain foundation models. We find that models pre-trained with longer contexts learn representations that transfer better to word decoding. Our results indicate that long-context pre-training helps exploit extended neural context that other methods unnecessarily discard.

[1]PNPL 🍍, University of Oxford. Correspondence to: Dulhan Jayalath <dulhan@robots.ox.ac.uk>.

*Proceedings of the 43$^{rd}$ International Conference on Machine Learning*, Seoul, South Korea. PMLR 306, 2026. Copyright 2026 by the author(s).

## 1. Introduction

Across modalities in deep learning, extending context has unlocked capabilities that short contexts could not provide. For example, task performance has improved by pre-training with longer, un-fragmented documents in language models (Dai et al., 2019; Beltagy et al., 2020), and by including more task examples in context at inference time (Brown et al., 2020). In audio modelling, dilated convolutions have extended the receptive field, capturing long-range structure that local windows miss (van den Oord et al., 2016). In video generation, models have struggled with temporal coherence when context is too short to maintain consistency across frames (Yan et al., 2023). The general pattern is that if a signal carries structure at timescale $T$, then models limited to a context shorter than $T$ cannot exploit it. We postulate that neural recordings during speech perception are no different. If the brain encodes speech-relevant information across extended timescales, especially if this content mirrors linguistic structure, then models with access to extended neural windows should recover information unavailable to short-context approaches.

Notwithstanding these predictions, prior brain-to-text decoders typically operate on brief samples of brain data in the range of milliseconds to seconds (Moses et al., 2021; Défossez et al., 2022; Tang et al., 2023; Willett et al., 2023; Card et al., 2024). This falls far short of the scale of linguistic context reflected by modern LLMs. While some neural

We have released code and model weights with instructions.

decoding systems independently incorporate longer linguistic context via external language models (Moses et al., 2021; Willett et al., 2023; Jayalath et al., 2025a), they nonetheless ignore the long-range context present in the neural data itself. Short context windows are chosen to broadly match the timescale of the unit being decoded (a phoneme, syllable, or word). This framing assumes that the relevant neural information is local: that decoding the current moment requires only the current signal. But brain recordings carry structure beyond the immediate stimulus. Subject-specific neural patterns, noise characteristics, scanner properties, and other factors can all shape the signal. In speech, phonemes group into syllables, syllables into words, and words into phrases, sentences, and discourses. Just as words in a sentence are interdependent, their corresponding neural representations are likely correlated. However, short contexts are too brief to capture and leverage these long-range dependencies.

Extending neural context has already shown promise in non-invasive decoding, where sensors sit outside the skull. While non-invasive approaches enable safe and scalable data collection, they have lower fidelity than surgical implants—a gap that longer context windows may help close. d'Ascoli et al. (2025) decoded words by modelling MEG signals at the sentence level, providing neural responses to all words in a perceived sentence to a transformer at once. Their work demonstrated that additional neural context could improve word decoding accuracy by 50% over isolated word classification. Despite this promising advance, reaching a reasonable accuracy still requires tens of hours of subject training data, with their analyses showing hours-per-subject matters more than total hours. Future clinical applications of speech *brain-computer interfaces (BCIs)* will need to work for patients with minimal labelled data because paralysed patients, who are the intended users, may not be able to easily provide training recordings (Willett et al., 2023; Card et al., 2024).

Pre-trained models should improve data efficiency by learning transferable representations across subjects, reducing the need for subject-specific training data. Yet existing methods (a) seem to only perform well in data-rich settings (Yang et al., 2026) and (b) rarely exploit long neural context because they are designed for tasks that seemingly require only short context. While Jayalath et al. (2025b) represents the only non-invasive pre-trained model tested on speech decoding, this model uses sample windows of just 0.5s as it was intended for simple speech sub-tasks where the stimulus—or the relevant neural response—does not span much longer than half of a second. Outside of speech, typical evaluation datasets include TUAB and TUEB (Obeid & Picone, 2016) where samples are often split to only 5 or 10 seconds long. As a result, foundation models like LaBraM (Jiang et al., 2024), BioCodec (Avramidis et al., 2025), and EEGPT (Wang et al., 2024) are all pre-trained with short samples of at most 10 seconds.

Leveraging long contexts is also limited by computational resources. BrainOmni (Xiao et al., 2025) and CBraMod (Wang et al., 2025) are the only non-specialised models to be trained with relatively long input contexts (up to 30s) avoiding the computational barrier of quadratic attention by factorising attention in time and sensors.

Overcoming these limitations presents a formidable challenge. Thus, to improve the data efficiency of brain-to-text while maintaining the advantages of extended neural context, we introduce MEG-XL, a long-context pre-training framework and model. The name pays homage to Transformer-XL (Dai et al., 2019), one of the first long-context transformer language models and where 'XL' stands for *extra long*. Accordingly, our method learns to model 2.5-minute long MEG samples, aiming to elicit both the advantages of improved decoding accuracy with long neural context, as shown by d'Ascoli et al. (2025), and the data efficiency afforded by long-context priors acquired through pre-training. These samples are 5-300× longer than prior pre-trained models, and correspond to 191k tokens[1]. This is similar to contexts in contemporary language models, e.g. GPT-4 Turbo (OpenAI, 2023), though the notion of a 'token' differs across domains. Using masked token prediction, MEG-XL learns from hundreds of subjects and hundreds of hours of recordings across rest, motor, speech, and other tasks. Like Xiao et al. (2025), it also overcomes the computational barrier of long-context modelling with memory-efficient criss-cross attention (Wang et al., 2025).

Fine-tuned on contextual word decoding across three MEG speech datasets, the model generalises to new subjects with limited data better than the state-of-the-art supervised approach (d'Ascoli et al., 2025). Therefore, pre-training successfully compensates for limited subject-specific data; with extensive 'deep' single-subject recordings, supervised models can eventually exploit rich in-domain patterns. Compared to brain foundation models, MEG-XL outperforms all alternatives in the 'shallow' low-data regime, where models have access to limited subject data, while remaining competitive when given deep subject recordings.

Performance may scale with the context length used for pre-training. With linear probing, we find that models pre-trained with longer neural contexts exhibit better representations for brain-to-text decoding. We also find evidence that long neural contexts may be beneficial beyond brain-to-text. Zero-shot prediction of masked brain signals from unseen datasets improves with models trained on longer neural context. Analysing attention patterns, we find this benefit stems from learning selective, hierarchical attention, with local processing in early layers expanding to global integration in later layers.

---

[1]Number of embeddings input to MEG-XL per sample, calculated via channels × timesteps × tokenizer compression ratio.

Our main contributions are as follows:

- **Long-Context: we scale neural pre-training to the minute-regime with contexts 5–300× longer than prior work.** MEG-XL models 2.5-minute MEG samples, allowing it to learn long-range dependencies in brain activity that short-window approaches cannot.

- **Data-Efficient Transfer: we address clinical cross-subject transfer by generalising with less data.** MEG-XL significantly outperforms state-of-the-art supervised methods and a comprehensive set of modern foundation models when subject-specific data is scarce.

- **Interpretation: the capacity to exploit long-range structure is a learned skill that improves with scale.** We provide empirical evidence that increasing pre-training context improves (a) brain-to-text and (b) zero-shot prediction of unseen data. This improvement stems from the emergence of selective, hierarchical attention patterns that short-context models never learn.

## 2. Method

We introduce MEG-XL, a self-supervised framework for learning representations from long-context MEG recordings (Figure 2). We adopt masked token prediction as our pre-training objective because it hides portions of the MEG signal and the model learns to reconstruct them from surrounding context. This directly trains the model to leverage extended temporal windows since predicting masked segments requires extracting information from whatever remains visible. Masked token prediction is also a well-tested approach, with success across many domains (Devlin et al., 2019; Baevski et al., 2020; He et al., 2022). As MEG signals are highly autocorrelated in time, our objective masks long blocks of tokens to avoid trivially interpolating from neighbouring time points and encourage learning non-trivial, longer range relationships from neural data.

To tokenize brain signals to serve as targets for this objective, our approach draws inspiration from generative audio models, particularly SoundStorm (Borsos et al., 2023), which use *residual vector quantization (RVQ)* in the tokenizer because residual tokens capture high-frequency time-series data with better fidelity. However, unlike brain models, audio models typically operate on single-channel waveforms while MEG recordings are multivariate, with hundreds of spatially distributed sensors. Thus, our framework adapts this method with sensor embeddings which have become standard in neural foundation models, e.g. Xiao et al. (2025).

Importantly, long neural contexts are tokenized into long spans of tokens. To efficiently model so many tokens without the computational bottleneck of quadratic self-attention (Vaswani et al., 2017), we use a criss-cross transformer (Wang et al., 2025), which provides an efficient factorisation of the transformer attention mechanism across time and channels.

### 2.1. Tokenizer

Raw MEG signals are first converted to discrete tokens using a frozen neural tokenizer. Given a multi-channel MEG recording $\mathbf{X} \in \mathbb{R}^{C \times T}$ with $C$ sensors and $T$ time samples, we apply BioCodec (Avramidis et al., 2025) independently to each channel. This tokenizer uses RVQ (Zeghidour et al., 2022; Défossez et al., 2023) with $Q = 6$ levels and vocabulary size $V = 256$, to produce codes $\mathbf{Z} \in \{0, \ldots, V - 1\}^{C \times T' \times Q}$ where $T' = T/r$ and $r = 12$ is the temporal downsampling factor of the tokenizer. RVQ captures both slow dynamics and high frequency patterns through hierarchical decomposition, allowing the codebook to have more capacity without exponential vocabulary growth (Zeghidour et al., 2022; Défossez et al., 2023; Avramidis et al., 2025).

Unlike most other work, which resample MEG recordings to 200-300Hz, we downsample to 50Hz because (a) this follows d'Ascoli et al. (2025) who introduced the word decoding task that we are interested in, and (b) it significantly reduces the number of tokens per sample, alleviating the memory bottleneck of modelling long contexts with attention. Although BioCodec was trained on EEG rather than MEG, and at 250Hz instead of 50Hz, we found it could reconstruct our data well (Appendix E). Alternative approaches, such as BrainTokenizer (Xiao et al., 2025), compress across channels in addition to time; this degraded reconstruction quality and therefore risked discarding more task-relevant information.

We note that the tokenization serves two purposes: it compresses the input by a factor of $r$, enabling the transformer to attend over longer temporal contexts, and it provides a natural prediction target for self-supervised learning with masked token prediction.

### 2.2. Model Architecture

**Input embeddings.** For each sensor $c \in C$ and timestep $t \in T$, we construct an embedding by (1) looking up the frozen codebook vectors at each RVQ level $q \in \{1, \ldots, Q\}$, (2) concatenating these vectors, and (3) projecting to the model dimension $d_{\text{model}}$:

$$\mathbf{h}_{c,t}^{(0)} = \mathbf{W}_{\text{proj}} \left[ \mathbf{e}_{z_{c,1,t}}^{(1)}; \ldots; \mathbf{e}_{z_{c,Q,t}}^{(Q)} \right] \tag{1}$$

where $\mathbf{e}_k^{(q)} \in \mathbb{R}^{d_{\text{codebook}}}$ denotes the $k$-th entry of the $q$-th codebook and $[\cdot; \cdot]$ denotes concatenation.

**Sensor embeddings.** MEG sensors vary in position, orientation, and type across recording systems. Following Xiao et al. (2025), we incorporate this structure through three

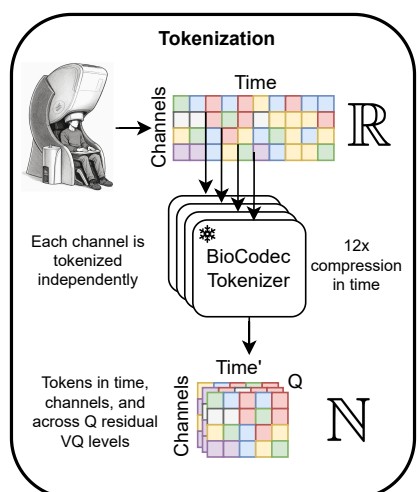
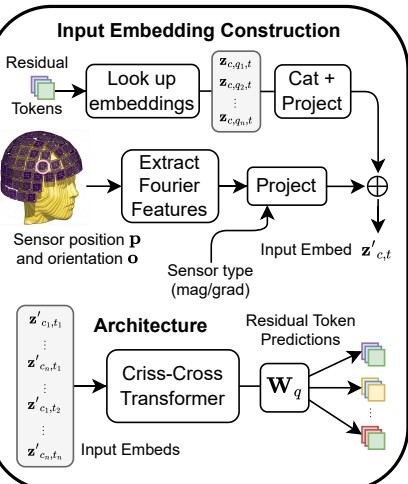
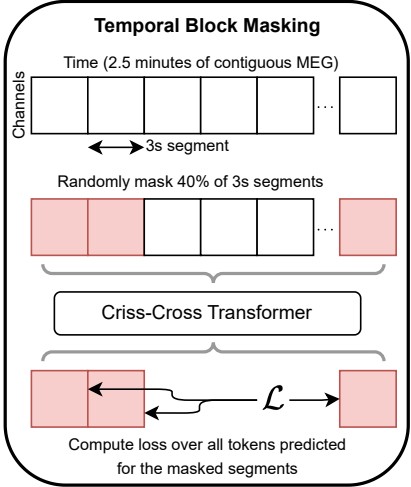

*Figure 2.* **Overview of the MEG-XL pre-training framework.** (Left) A frozen BioCodec tokenizer independently encodes each MEG channel into discrete tokens across Q=6 residual quantization levels, providing prediction targets for self-supervised learning. (Middle) Token embeddings, which are concatenated across quantization levels and projected, are combined with sensor position, orientation, and type embeddings, then processed by a criss-cross transformer. A projection head maps transformer embeddings back to Q residual tokens. (Right) In pre-training, we randomly mask contiguous 3-second blocks uniformly across all sensors, forcing the model to predict masked tokens from temporal context rather than interpolating across channels, until 40% of tokens are masked.

additive embeddings. Position and orientation are encoded via Gaussian Fourier features (Tancik et al., 2020):

$$\gamma(\mathbf{v}) = [\cos(2\pi\mathbf{B}\mathbf{v}), \sin(2\pi\mathbf{B}\mathbf{v})] \quad (2)$$

where $\mathbf{B} \in \mathbb{R}^{d_{\text{fourier}}/2 \times 3}$ contains frequencies sampled from $\mathcal{N}(0, \sigma^2)$. Separate Fourier embeddings encode sensor position $\mathbf{p}_c \in \mathbb{R}^3$ and orientation $\mathbf{o}_c \in \mathbb{R}^3$. Sensor type (gradiometer or magnetometer) is encoded via a learned embedding. All three are projected to the transformer dimension $d_{\text{model}}$ and summed with the token embedding.

**Architecture.** The transformer consists of $L = 8$ layers and each transformer block follows a pre-layer normalisation architecture: RMSNorm (Zhang & Sennrich, 2019) precedes the criss-cross attention and feedforward sub-layers, with residual connections applied after each. The feedforward network uses 4× hidden expansion with SELU activation (Klambauer et al., 2017). The criss-cross attention modules (Wang et al., 2025; Xiao et al., 2025) split the feature dimension in half and apply temporal and spatial attention in parallel to the respective halves:

$$\mathbf{H}^{(\ell)} = \mathbf{H}^{(\ell-1)} + \Big[\text{SpatialAttn}(\mathbf{H}^{(\ell-1)}_{:d_{\text{model}}/2});$$
$$\text{TemporalAttn}(\mathbf{H}^{(\ell-1)}_{d_{\text{model}}/2:})\Big] \quad (3)$$

where $[\cdot; \cdot]$ denotes concatenation along the feature dimension. Temporal attention operates independently per sensor across the time axis, while spatial attention operates independently per timestep across sensors. Rotary position embeddings (Su et al., 2024) encode temporal position within temporal attention blocks. The factorization to temporal and

spatial attention reduces the quadratic cost of full attention from $\mathcal{O}((CT')^2)$ to $\mathcal{O}(C \cdot T'^2 + T' \cdot C^2)$. We find this essential for modelling long context neural data, especially as our tokenizer does not compress the sensor channel dimension.

### 2.3. Masked Prediction Objective

We train the model to predict masked tokens from surrounding context in 2.5-minute long samples. We use *temporal block masking*, masking randomly selected 3-second blocks until we mask 40% of the sequence, where the masking percentage was determined through manual tuning. Interestingly, this is below the 75% used on images in masked autoencoders (He et al., 2022), far above the 15% used by BERT for language modelling (Devlin et al., 2019), but close to the 49% in wav2vec 2.0 (Baevski et al., 2020), a self-supervised auditory speech model. We use large masking blocks of 3s, rather than shorter periods, because it forces the network to model neural patterns across long periods of time. It also covers the length of decodable neural responses to words (Kutas & Federmeier, 2011; Fyshe et al., 2019). The mask is applied uniformly across all sensors at the selected timesteps, preventing the model from only interpolating across simultaneous sensor readings. Masked tokens are replaced with a learnable mask embedding.

Let $\mathcal{M} \subset \{1, \ldots, T'\}$ denote the set of masked timesteps. For each masked position, the model predicts the discrete code at each RVQ level via a linear head:

$$p(z_{c,q,t} \mid \mathbf{X}_{\setminus\mathcal{M}}) = \text{softmax}(\mathbf{W}_q \mathbf{h}^{(L)}_{c,t}) \quad (4)$$

The training objective is the average cross-entropy across

*Table 1.* **Comparison to Foundation Models.** We end-to-end fine-tune all models for word decoding, with individual details on fine-tuning provided in Appendix A.3. The quoted uncertainty is the standard error of the mean over three seeds. Best results are indicated in **bold** and second-best in ***bold-italic***. Results with * indicate the best result beats the second-best with $p < .05$ (with Welch's $t$-test).

| Model | Params. | With 13% of training data | | | With 100% of training data | | |
| --- | --- | --- | --- | --- | --- | --- | --- |
| | | MEG-MASC | Armeni | LibriBrain | MEG-MASC | Armeni | LibriBrain |
| BioCodec | 1.0M | $19.8 \pm 0.3$ | $20.0 \pm 0.2$ | $19.9 \pm 0.1$ | $31.2 \pm 2.1$ | $37.1 \pm 0.8$ | $41.9 \pm 1.0$ |
| EEGPT | 4.7M | $19.6 \pm 0.5$ | $20.3 \pm 0.1$ | $20.3 \pm 0.4$ | $26.3 \pm 2.2$ | $20.8 \pm 0.2$ | $22.9 \pm 0.3$ |
| BIOT | 3.2M | $20.0 \pm 2.2$ | $20.2 \pm 0.2$ | $20.6 \pm 0.3$ | $31.3 \pm 1.7$ | $35.7 \pm 4.4$ | $45.6 \pm 0.4$ |
| BBL | 15M | $21.5 \pm 0.5$ | $22.3 \pm 0.4$ | $32.1 \pm 1.2$ | ***$35.9 \pm 1.2$*** | $39.1 \pm 0.2$ | $49.9 \pm 0.3$ |
| BrainOmni | 8.4M | $18.7 \pm 0.7$ | $21.0 \pm 1.2$ | $29.7 \pm 9.3$ | $19.1 \pm 2.8$ | **$62.3 \pm 0.8$** | **$63.0 \pm 0.1$** |
| LaBraM | 5.8M | ***$33.2 \pm 1.0$*** | ***$26.3 \pm 2.4$*** | ***$40.3 \pm 0.1$*** | $31.1 \pm 1.0$ | $42.0 \pm 0.5$ | $47.7 \pm 0.3$ |
| **MEG-XL (Ours)** | 20M | **$47.0 \pm 0.9$*** | **$54.9 \pm 0.5$*** | **$57.3 \pm 0.4$*** | **$46.4 \pm 1.3$*** | ***$61.2 \pm 0.4$*** | **$63.0 \pm 0.4$** |

masked positions and RVQ levels:

$$\mathcal{L} = -\frac{1}{|\mathcal{M}| \cdot C \cdot Q} \sum_{t \in \mathcal{M}} \sum_{c=1}^{C} \sum_{q=1}^{Q} \log p(z_{c,t,q} \mid \mathbf{X}_{\backslash \mathcal{M}}) \quad (5)$$

We note that different MEG systems have different sensor counts. We pad recordings to a maximum channel count and we use a sensor mask to exclude padded channels from both attention computations and the loss. This enables training on heterogeneous datasets without architectural changes.

## 3. Experiments

Our experiments evaluate our pre-trained approach on word decoding by fine-tuning it end-to-end on speech-task MEG datasets. We compare it to the state-of-the-art supervised method in contextual word decoding (d'Ascoli et al., 2025) as well as several recent brain foundation models across data regimes. We then investigate generalisation when models are pre-trained with increasing neural context.

**Pre-training data.** We pre-train on approximately 300 hours of MEG data compiled from the CamCAN (Shafto et al., 2014), MOUS (Schoffelen et al., 2019), and SMN4Lang (Wang et al., 2022) datasets (see Appendix A.2)

**Contextual word decoding task.** We follow d'Ascoli et al. (2025)'s word-locked epoch decoding strategy. Given a sequence of $N = 50$ words, we construct the input by extracting 3-second neural windows $\mathbf{X}_n$, aligned 0.5s before word onset, and concatenate them along the time axis. This results in a single input tensor of 150s (50 words $\times$ 3s), denoted as $\{\mathbf{X}_n\}_{n=1}^{N}$. This approach (a) strictly replicates the supervised baseline for fair comparison, and (b) partially removes the confound of variable speech rates while preserving the temporal evolution of each word's neural response. Although this input is not perfectly continuous in physiological time, it still aligns with our pre-training strategy as the 150-second context window in pre-training covers the duration of the 50-word sequence in fine-tuning. This allows our

model to leverage its learned long-range priors without architectural modification. The model is trained to predict semantic target word embeddings $\{\mathbf{w}_n\}_{n=1}^{N}$ extracted from a T5 large language model (Raffel et al., 2020) using a contrastive SigLIP variant (Zhai et al., 2023) that masks repeated words in a sequence. For each word $n$, we extract transformer outputs from the last layer $\mathbf{h}_{c,t}^{(L)}$ over the corresponding time interval, average across time, and flatten across channels to obtain $\mathbf{r}_n \in \mathbb{R}^{Cd}$. An MLP head maps this to the predicted embedding: $\hat{\mathbf{w}}_n = \text{MLP}(\mathbf{r}_n)$. We fine-tune the transformer and MLP head end-to-end. At inference, words are predicted by nearest-neighbor retrieval, maximising cosine similarity: $\hat{y}_n = \arg\max_y \cos(\hat{\mathbf{w}}_n, \mathbf{w}_y)$.

**Evaluation data.** We evaluate with the three largest English-language perceived speech MEG datasets (Table 2). These are LibriBrain (Özdogan et al., 2025; Landau et al., 2025), a deep dataset of 52 hours of MEG from a single subject listening to audiobooks, Armeni et al. (2022), in which 3 subjects each listened to 10 hours of audiobooks, and MEG-MASC (Gwilliams et al., 2023), a broad study with 27 subjects listening to 2 hours each of short stories. We split sequences into train, validation, and test splits in the ratio 80:10:10. The same stimuli presented to different subjects are always assigned to the same set, preventing the model from exploiting stimulus-specific features on held-out subjects (Jo et al., 2024).

*Table 2.* **Evaluation datasets.** The three datasets span different regimes: shallow multi-subject (MEG-MASC), deep single-subject (LibriBrain), and somewhere in-between (Armeni).

| Dataset | Subjects | Hrs./Subject | Total Hrs. |
| --- | --- | --- | --- |
| MEG-MASC | 27 | 2 | 54 |
| Armeni | 3 | 10 | 30 |
| LibriBrain | 1 | 52 | 52 |

**Metric.** Following d'Ascoli et al. (2025), we measure word decoding performance with top-10 balanced accuracy over a fixed retrieval vocabulary. This is the macro average

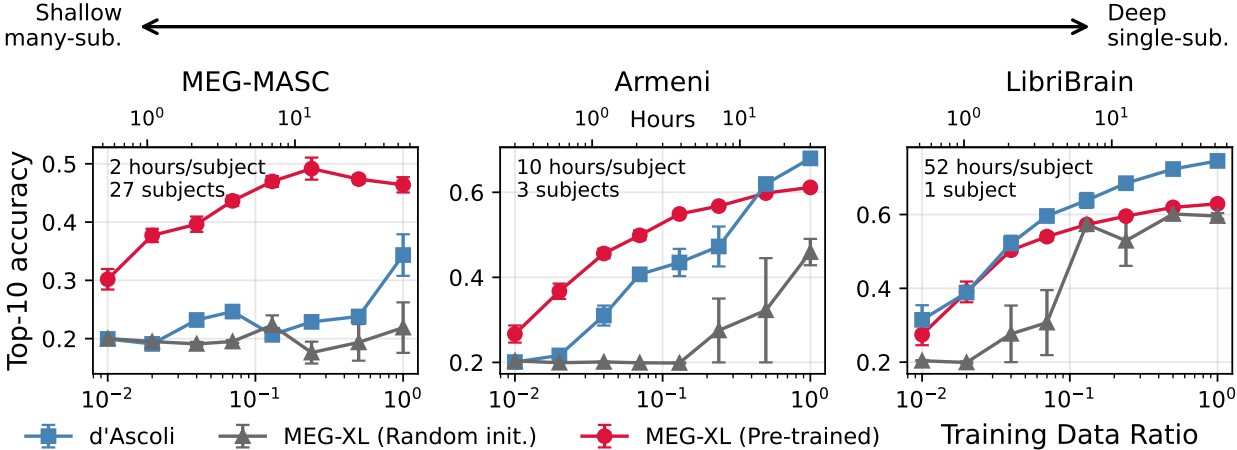

*Figure 3.* **Pre-training enables generalisation with less subject data.** We compare MEG-XL to the state-of-the-art supervised method (d'Ascoli et al., 2025) and a baseline trained from scratch (MEG-XL with random init.) across varying amounts of fine-tuning data. MEG-XL consistently outperforms its randomly initialised counterpart, confirming that the gains stem from learned priors rather than architecture alone. On Armeni and MEG-MASC, where per-subject data is shallower, MEG-XL outperforms d'Ascoli et al. (2025) throughout most of the data range. On LibriBrain, a deep single-subject dataset, both methods perform similarly until approximately 2.5 hours of training data, after which their supervised method pulls ahead. This suggests pre-training may substitute for subject-specific data as when recordings are scarce, learned priors help. In rare cases where recordings are abundant, learning from scratch eventually wins.

of top-10 accuracy over word classes, accounting for word frequency imbalance. d'Ascoli et al. (2025) use retrieval sets of the top-50 and top-250 most frequent words. For consistency, we use top-50; top-250 results in Appendix B show similar trends. We report this metric on the test set using the best validation checkpoint (early stopping).

### 3.1. Comparison to Foundation Models

Comparing MEG-XL to six state-of-the-art pre-trained models, our approach shows the best generalisation under data constraints and joint best with all data. Table 1 reports performance at 13% and 100% of training data. Here, we randomly subsample the aggregate training set. Since our multi-subject datasets are balanced, these ratios correspond to a proportional reduction in data per subject. Below 13%, most baselines collapse to chance performance, making comparison uninformative; this threshold thus represents the low-data regime where foundation models can plausibly compete. With limited data, MEG-XL outperforms all baselines by substantial margins. With full data, MEG-XL matches or exceeds all methods on LibriBrain and MEG-MASC, while remaining within 1.1 percent of BrainOmni on Armeni. Notably, BrainOmni's strong performance on LibriBrain and Armeni does not transfer to MEG-MASC, suggesting that this model does not perform well with shallow per-subject data. This is the regime that matters for clinical deployment as transfer to new users needs models that generalise from little subject data.

MEG-XL is most effective precisely in the data-constrained regime where other foundation models collapse. While re-

cent analysis suggests pre-trained models struggle with limited data (Yang et al., 2026)—a trend confirmed by our baselines, which largely remain at chance levels with 13% training data—MEG-XL breaks this pattern. It achieves 47–57% accuracy against a 20% random baseline, significantly outperforming the only other viable alternative, LaBraM. Thus, MEG-XL's pre-training may capture structure that helps generalise with greater data efficiency.

### 3.2. Data Efficiency Compared to Supervised Learning

Evaluations of brain foundation models often lack supervised baselines trained on equivalent data, obscuring the true value of pre-training. To address this, we benchmark MEG-XL directly against the state-of-the-art supervised method (d'Ascoli et al., 2025) across the full data spectrum.

Our approach performs well in the low-data regime, generally outperforming d'Ascoli et al. (2025) (Figure 3), except on deep single-subject data. The significant gap between the pre-trained and randomly initialised MEG-XL models demonstrates that the architecture itself is insufficient without the learned priors acquired during pre-training. Against d'Ascoli et al. (2025), on MEG-MASC, our model improves over 25% at certain points and the benefits persist across all 54 hours of training data (2 hours per subject). On Armeni et al. (2022), fine-tuning our model leads to accuracy roughly 10% above the supervised method until around 15 hours of data (5 hours per subject). Conversely, on LibriBrain both methods perform similarly until 2.5 hours, after which the d'Ascoli et al. (2025) method pulls ahead. Pre-training helps most with more subjects and when per-subject

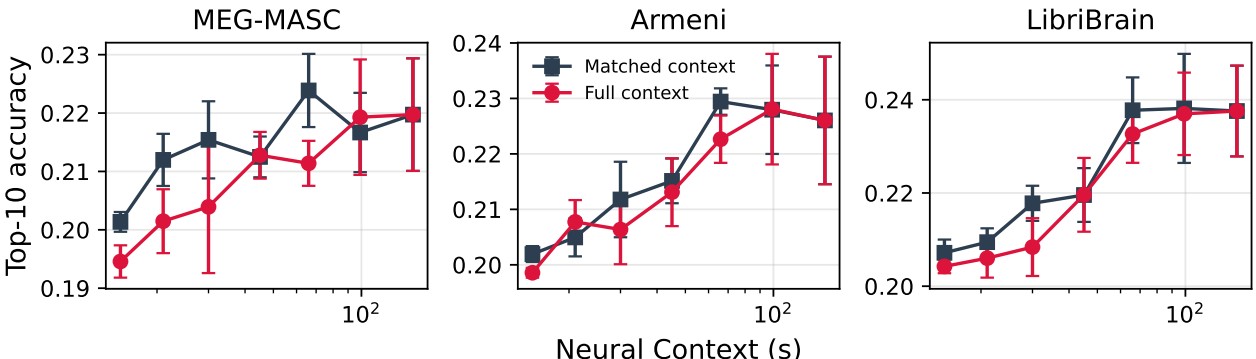

*Figure 4.* **Linear probing shows that models pre-trained with more context generalise better to word decoding.** We pre-train models with increasing context, fixed masking percentage, and constant optimisation steps, then evaluate the strength of their representations with linear probes (frozen backbone). We compare two conditions: *full context*, where all models see 150s of input to isolate representation quality, and *matched context*, where the input is restricted to the pre-training length. The lack of divergence between the lines suggests models cannot leverage inference context that exceeds their pre-training context. We train the linear probes with 7% of the training data. We could not expand further than 150s due to GPU memory limits. Token-matched pre-training shows similar trends (Appendix D).

data is shallow, but deep single-subject recordings eventually favour learning from scratch. These results suggest one relevant axis of scale for clinical neural decoding may be breadth across subjects. When per-subject data is constrained, as with paralysed patients, pre-training on many subjects substitutes for extensive individual training.

Recent work has questioned whether brain foundation models improve over supervised baselines, finding marginal gains of 1-2% despite orders of magnitude more parameters (Lee et al., 2025b), and poor performance in low-data regimes (Yang et al., 2026). In contrast, our results show 10–25% gains over the supervised state-of-the-art (d'Ascoli et al., 2025), with fewer parameters (20M vs 200M), specifically in the low-data regime that is most important for clinical deployment. The difference may lie in alignment between pre-training and downstream task. Foundation models assume generality, expecting representations to transfer broadly. However, speech decoding may demand temporal structure that short-context pre-training discards. Domain-aligned pre-training with extended context outperforms more generic models trained with more data. The optimal pre-training strategy depends not only on scale but on the temporal and structural demands of the target task.

### 3.3. Longer Context Improves Representations

To what extent can we exploit neural context in pre-training? We pre-train models with increasing context length, freeze their weights, and train linear probes on the final-layer representations to predict word embeddings. Figure 4 shows that longer pre-training context yields better word decoding across datasets, with diminishing returns after one hundred seconds. With full context, differences reflect representation quality, not inference-time context; with matched context, we ensure that differences are not due to mismatch between

pre-training and probing context. Moreover, the small difference between full and matched context probing suggests that additional inference-time context provides no benefit unless the model was pre-trained to use it. This mirrors the 'length generalisation' challenge in large language models, where performance typically plateaus when inference context exceeds the pre-training context (Dai et al., 2019; Press et al., 2022), confirming that long-context utilisation is also a learned capability in the neural data domain.

Could the benefit of neural context extend beyond brain-to-text? Figure 5 (top) shows that extending neural context during pre-training consistently improves zero-shot masked token prediction across all three unseen datasets. This scaling appears log-linear, indicating that longer context improves modelling of neural data. Notably, while linear probing for speech decoding plateaus after 100 seconds, masked prediction does not. This discrepancy suggests that the utility of context is task-dependent, as not all information captured in the neural signal is necessary for every downstream task.

### 3.4. What Does Long-Context Pretraining Teach?

We analyse temporal attention patterns across models pre-trained with varying context lengths. Short-context models attend diffusely from the first layer, spreading attention uniformly across time. Longer-context models attend locally in early layers, then progressively expand their temporal reach through depth (Figure 5 bottom left). This hierarchical processing coincides with lower attention entropy. Thus, longer-context models are more selective about which timesteps matter (Figure 5 bottom right). These results suggest long-context pretraining teaches models when to attend far versus near. Together with Figure 5 (top), this also provides further evidence that long-context data can only be properly leveraged with long-context pre-training.

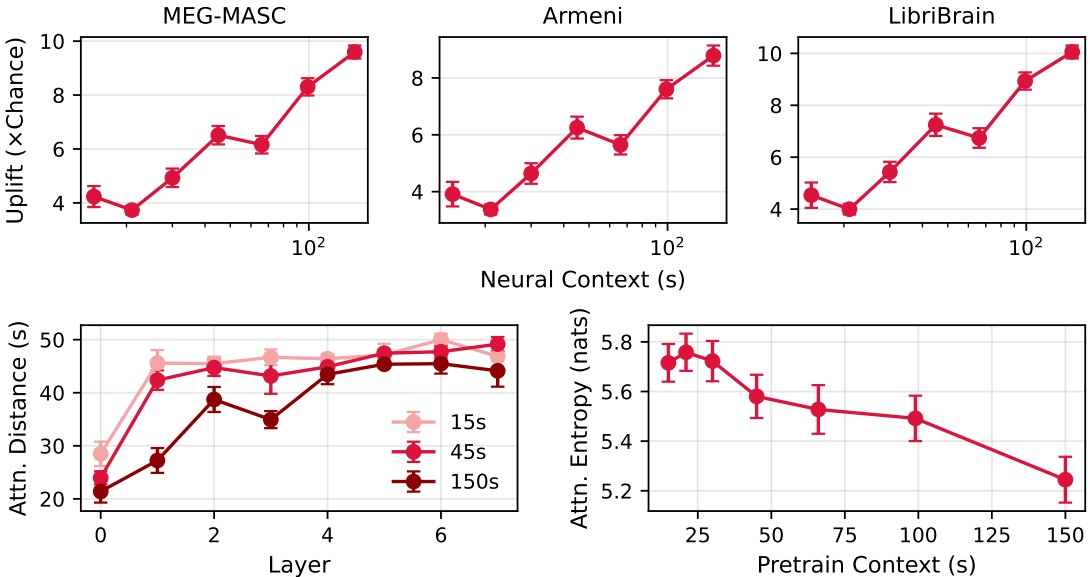

*Figure 5.* **(Top) Extending neural context improves zero-shot prediction of brain activity from unseen datasets and subjects.** We mask the central 3s subsegment of samples from unseen datasets and measure improvement in token prediction accuracy (relative to chance) of models pre-trained on increasing neural context. Scaling improves masked prediction, with the trend remaining through 150s. Only GPU VRAM limits prevent increasing it further. Chance accuracy is 1/256. **(Bottom) Long-context pretraining induces selective and hierarchical attention.** (Left) Models pretrained on longer context attend locally in early layers before expanding to integrate distant context; short-context models attend diffusely throughout. (Right) Attention entropy decreases with context length, indicating more selective attention patterns. We provide 150s context at inference. See Appendix F for attention distance and entropy calculations.

## 4. Discussion

Supervised training on large within-subject data remains the gold standard and is hard to beat—but this currently limits applicability to new subjects such as paralysed patients who cannot provide long training recordings. We show that pre-training with long-context reduces the data requirement from dozens to 1-2 hours of total data and only tens of minutes per subject, vastly expanding the range of applicable subjects. Thus, although clinical deployment remains distant, requiring extension to imagined speech and patient populations, the subject-specific data bottleneck may be softer than assumed. Pre-training on long neural context across subjects can substitute for extensive per-subject data.

These improvements in data efficiency are driven by scaling neural context in pre-training. As we show, representations continue to generalise better in word decoding, with diminishing returns after 100s of context. Models pre-trained on longer windows also improve at zero-shot prediction of masked brain activity, where performance scales to 150s without saturation. This may suggest that context could help on other tasks, though here we focus on word decoding. Whether the asymmetry between tasks reflects task-specific ceilings or linear readout limits remains unclear. Further scaling may benefit some tasks more than others.

The reason long-context pre-training helps is that it teaches models to attend more selectively and to hierarchically process near-to-far context. This is an ability that short-context

models never acquire. Thus, instead of unlocking the ability to attend to far context, the main advantage lies in learning when and how to make use of distant context.

These findings come with some caveats. We focus on perceived speech as a foundational step before addressing imagined speech. We also evaluate on a relatively limited 50-word vocabulary. Performance decreases with larger vocabularies (Appendix B), though this parallels the development of open-vocabulary decoding in surgical studies, where results were first obtained on 50-word vocabularies before scaling up to hundreds of thousands of words (Moses et al., 2021; Willett et al., 2023). While we characterise the attention patterns that drive generalisation, understanding the nature of the learned structure also remains unresolved.

To conclude, neural decoders have historically operated on windows shorter than the structure they aim to recover. While supervised approaches have begun to widen this window, e.g. d'Ascoli et al. (2025), providing long contexts to a model is insufficient for data-efficient generalisation without appropriate long-context statistical priors. Just as language models require pre-training on long documents to master coherence, brain-to-text systems benefit from pre-training with long-context neural activity to decode speech with limited subject data. By pre-training on extended contexts, we can now move beyond accessing long windows to acquiring the long-context priors to exploit them, recovering usable information that both short-context pre-trained models and purely supervised methods leave behind in data-scarce regimes.

## Impact Statement

This paper presents work towards non-invasive speech decoding with potential applications to individuals who have lost the ability to speak. Clinical deployment remains distant as performance is below what communication aids require and substantial work remains. Neural decoding technologies raise obvious privacy concerns as they involve inferring mental content from brain activity. The present work uses only publicly available research datasets with their own ethics approvals and decodes perceived speech rather than covert thought. As capabilities improve, the field will need norms around consent, data ownership, and the boundary between assistive and surveillant applications. These are questions we do not resolve here but consider essential.

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

# A. Details on Experimental Setup

## A.1. Preprocessing

We follow a minimal preprocessing pipeline similar to Défossez et al. (2022) and d'Ascoli et al. (2025). We preprocess all recordings with a 0.1Hz high-pass and 40Hz low-pass filter and then resample the recording to 50Hz. Although technically this risks creating aliasing artefacts, it follows the standard word decoding preprocessing pipeline in d'Ascoli et al. (2025) and reduces the number of timepoints, allowing for longer contexts. Appendix C contains results with Nyquist-compliant resampling. We standardise each contiguous 3s long subsegment within a 2.5-minute sample independently. We first baseline correct each subsegment by subtracting the channel-wise mean of the first 0.5s, then we scale such that the subsegment has median zero and upper quartile and lower quartile of 1 and -1 respectively. Standardising by subsegment makes the pre-training data distribution more representative of that seen in the downstream word decoding task. Finally, we clamp the signal to the range $(-5, 5)$.

While a 0.1Hz high-pass filter attenuates signals with periods longer than 10 seconds, i.e. "slow drift", this does not prevent modelling structure at longer timescales. The frequency content of the signal and the timescale of dependencies in that signal are distinct. The high-pass filter removes slow-drift and scanner artifacts. It does not remove the statistical dependencies between neural responses separated by minutes. A model with access to 2.5 minutes of (filtered) signal can still learn these dependencies while a model with access to 2 seconds cannot.

## A.2. Pre-training Datasets

We use continuous 2.5-minute windows of MEG data as samples from the following datasets: CamCAN (Shafto et al., 2014), a healthy ageing study with rest, sensorimotor, and passive sensory task data from approximately 700 subjects, MOUS (Schoffelen et al., 2019), a 104-subject language study with data from reading and listening in Dutch, and SMN4Lang (Wang et al., 2022), another language study with 12 subjects listening to extended natural speech in Chinese (Mandarin). For CamCAN, we use only the rest and sensorimotor task subset, for MOUS, we use only the listening task data, and for SMN4Lang we use all the available MEG data. This totals approximately 300 hours of pre-training data and over 800 subjects.

## A.3. Foundation Model Baselines

For each baseline, unless otherwise specified, we provide full 2.5-minute long samples and fine-tune pre-trained checkpoints end-to-end alongside an MLP head (with hidden dimension 2048) that predicts the target embedding. The output embeddings from model backbones are sliced according to their alignment with word stimuli and pooled in the time dimension before being flattened and concatenated. These new embeddings, each corresponding to a word, are given independently to the MLP head. We use a higher learning rate for the MLP, which is trained from scratch, just like we do with MEG-XL (see the hyperparameters in Table 3). We provide each foundation model with data of the same sample rate as used for pre-training.

**BioCodec.** Avramidis et al. (2025) trained BioCodec as a single-channel tokenizer across thousands of hours of EEG data. When evaluating BioCodec's generalisation, we treat the tokenizer as a pre-trained foundation model applied to each sensor channel independently. To resolve spatial features across channels, we follow the authors' suggestion of processing BioCodec embeddings with two linear transformers over time and sensors. We then pool embeddings in the time dimension before feeding the result to the two-layer MLP.

**BIOT.** Yang et al. (2023) trained BIOT as a foundation model for generic biosignals, but used primarily EEG data in pre-training. As the model was pre-trained with a maximum of 18 channels of data and MEG has many more channels, we insert a randomly initialised trainable projection before BIOT that reduces the channel dimension to comply. As the learnable positional embeddings in the model have a maximum trained length, we had to reduce the neural context to 24 seconds for this model. We used the `EEG-six-datasets-18-channels` checkpoint that was trained on the most data.

**EEGPT.** EEGPT (Wang et al., 2024) is another pre-trained foundation model designed for generalising from EEG data. This model was pre-trained to support at most 58 channels. To support MEG data with many more channels, we add a randomly initialised trainable projection before the network to reduce the channel dimension of our MEG data to comply.

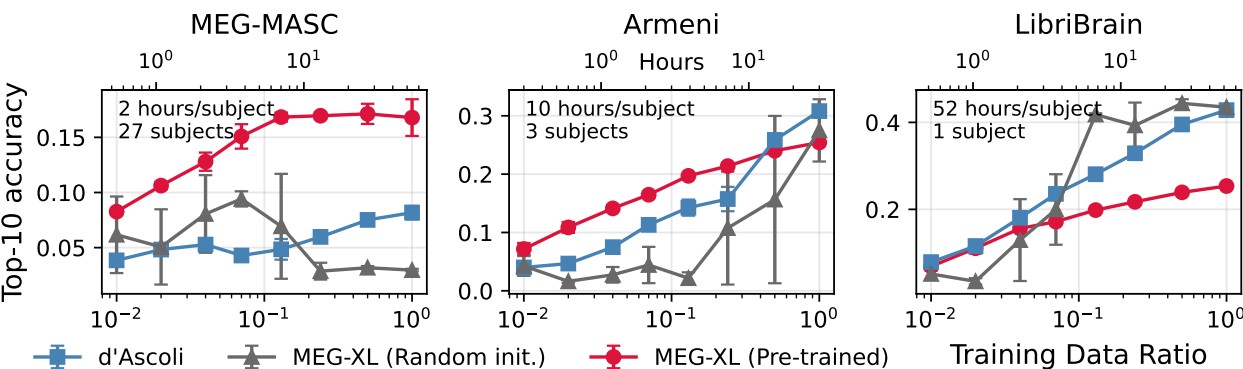

*Figure 6.* **Generalisation Across Training Data Regimes (Top 250 Words as Retrieval Set).** Results in the main paper use top-50 word retrieval sets. As expected, trends with top-250 word retrieval remain the same with the larger vocabulary leading to degraded performance across all methods. See the caption in Figure 3 for details.

**BBL.** *The Brain's Bitter Lesson* (Jayalath et al., 2025b) developed a model pre-trained on MEG data and designed for simple speech decoding tasks (speech detection, phonetic feature classification).

**BrainOmni.** Xiao et al. (2025) trained BrainOmni with a mix of both MEG and EEG data. The model's tokenizer leverages sensor position, orientation, and type. As our approach also uses this information, we provide it directly to BrainOmni with our datasets.

**LaBraM.** LaBraM (Jiang et al., 2024) is a large-scale EEG foundation model trained with a masked patch prediction objective. As LaBraM's learned time embeddings limit its context length, we had to reduce the neural context to 15 seconds.

### A.4. Supervised Word Decoding Baseline

To collect experimental results for d'Ascoli et al. (2025), we ran the code released as part of the supplementary materials of their publication, following the instructions provided in their README files. They did not support a data loader for LibriBrain, since this dataset was not evaluated in their work. Therefore, we implemented our own for evaluation.

### A.5. Hyperparameters

We provide all hyperparameters in Table 3.

### A.6. Computational Resources

All experiments were performed on individual NVIDIA A100, L40S, or H100 GPUs. Pre-training MEG-XL took approximately 12 hours on one H100. Fine-tuning MEG-XL with 30-50 hours of data took a similar amount of time.

## B. Larger Retrieval Set Results

For posterity, we provide the same comparison results as in the main body of the paper, except with the top 250 most frequent words as the retrieval set instead of the top 50 words. These are provided in Figure 6 and Table 4.

## C. Nyquist-Compliant Resampling

To match the preprocessing in d'Ascoli et al. (2025), we applied a 40Hz low-pass filter before resampling the brain data to 50Hz. This made it possible to do a like-for-like comparison with d'Ascoli et al. (2025). However, this technically violates a tenet of signal processing—specifically, the Nyquist criterion requires the sample rate to be at least twice the low-pass filter cut-off. To flout this risks introducing aliasing artefacts. In this appendix, we therefore repeat experiments from the main text but with 100Hz resampling. As this doubles the number of timepoints per sample, we reduced the context length from

*Table 3.* Hyperparameters. We use the implementation of criss-cross attention provided by Xiao et al. (2025) in their source code.

| Parameter | Value |
|---|---|
| **Data** | |
| Original sampling rate | 1000Hz |
| Re-sampled rate | 50Hz |
| Low-pass filter | 40Hz |
| High-pass filter | 0.1Hz |
| Standardization | IQR = [-1, 1] / 3s subsegment |
| Baseline correction | [0.0s, 0.5s] / 3s subsegment |
| Clamping range | [-5, 5] |
| Sample length | 150s |
| **Model** | |
| Gaussian Fourier Features (GFF) embedding dim. ($d_{\text{fourier}}$) | 256 |
| GFF sensor position $\sigma$ | 1.8 |
| GFF sensor orientation $\sigma$ | 1.0 |
| Transformer dim. ($d_{\text{model}}$) | 512 |
| Transformer layers | 8 |
| Transformer heads | 8 |
| Transformer attention | Criss-cross attention (Wang et al., 2025) |
| Mask block duration | 3s |
| Number of blocks masked | 20 ($\sim$40% of blocks) |
| **Pre-Training** | |
| Training steps | 35000 |
| Batch size | 1 |
| Learning rate | 1e-4 |
| Weight decay | 1e-4 |
| Warmup steps | 250 |
| Gradient clipping | 1.0 |
| Optimizer | AdamW (Loshchilov & Hutter, 2019) |
| Loss | Cross-entropy on masked tokens |
| **Fine-Tuning** | |
| MLP head hidden dim. | 2048 |
| Training steps | 50 epochs (max. with early stopping) |
| Early stopping patience | 10 epochs |
| Early stopping metric | Top-10 balanced accuracy on val. |
| Batch size | 50 words |
| Learning rate (transformer) | 1e-5 |
| Learning rate (MLP head) | 1e-3 |
| Weight decay | 1e-4 |
| Gradient clipping | 1.0 |
| Optimizer | AdamW (Loshchilov & Hutter, 2019) |
| Loss | D-SigLIP (Zhai et al., 2023; d'Ascoli et al., 2025) |

*Table 4.* **Comparison to Foundation Models (Top 250 Words as Retrieval Set).** See the caption in Table 1 for details.

| Model | Params. | With 13% of training data | | | With 100% of training data | | |
|---|---|---|---|---|---|---|---|
| | | MEG-MASC | Armeni | LibriBrain | MEG-MASC | Armeni | LibriBrain |
| BioCodec | 1.0M | $4.7 \pm 0.1$ | $4.0 \pm 0.3$ | $4.0 \pm 0.1$ | $10.1 \pm 0.5$ | $10.6 \pm 0.3$ | $12.4 \pm 0.4$ |
| EEGPT | 4.7M | $4.4 \pm 0.1$ | $4.0 \pm 0.2$ | $3.9 \pm 0.2$ | $6.7 \pm 0.5$ | $4.0 \pm 0.1$ | $5.1 \pm 0.1$ |
| BIOT | 3.2M | $3.5 \pm 0.7$ | $4.0 \pm 0.1$ | $4.2 \pm 0.0$ | $8.9 \pm 0.4$ | $9.9 \pm 1.8$ | $13.3 \pm 0.2$ |
| BBL | 15M | $3.8 \pm 0.6$ | $4.7 \pm 0.2$ | $7.7 \pm 0.5$ | $\boldsymbol{9.2 \pm 0.6}$ | $10.2 \pm 0.2$ | $16.1 \pm 0.2$ |
| BrainOmni | 8.4M | $3.7 \pm 0.8$ | $4.7 \pm 0.5$ | $7.6 \pm 3.5$ | $4.1 \pm 0.9$ | $\boldsymbol{26.8 \pm 0.5}$ | $\boldsymbol{25.6 \pm 0.1}$ |
| LaBraM | 5.8M | $\boldsymbol{9.9 \pm 0.6}$ | $\boldsymbol{6.5 \pm 0.9}$ | $\boldsymbol{12.0 \pm 0.3}$ | $8.9 \pm 1.1$ | $14.1 \pm 0.5$ | $16.9 \pm 0.2$ |
| **MEG-XL (Ours)** | 20M | $\boldsymbol{16.8 \pm 0.4}$ | $\boldsymbol{19.7 \pm 0.3}$ | $\boldsymbol{19.8 \pm 0.4}$ | $\boldsymbol{16.8 \pm 1.7}$ | $\boldsymbol{25.4 \pm 0.1}$ | $\boldsymbol{25.4 \pm 0.3}$ |

*Table 5.* **Nyquist-compliant resampling results.** Resampling to 100Hz avoids aliasing by respecting the Nyquist criterion (resampling frequency $\geq 2\times$ low-pass filter frequency). However, it requires reducing context length to 75s due to GPU VRAM constraints.

| Model | With 13% of training data | | | With 100% of training data | | |
|---|---|---|---|---|---|---|
| | MEG-MASC | Armeni | LibriBrain | MEG-MASC | Armeni | LibriBrain |
| MEG-XL (50Hz) | $47.0 \pm 0.9$ | $54.9 \pm 0.5$ | $57.3 \pm 0.4$ | $46.4 \pm 1.3$ | $61.2 \pm 0.4$ | $63.0 \pm 0.4$ |
| MEG-XL (100Hz) | $41.9 \pm 1.4$ | $51.7 \pm 1.0$ | $52.2 \pm 0.4$ | $41.8 \pm 2.7$ | $56.7 \pm 1.2$ | $58.6 \pm 0.1$ |

150s to 75s to satisfy GPU memory constraints. Table 5 compares the two configurations.

## D. Token-Matched Neural Context Scaling

Evaluating the effect of neural context length is complicated by confounds during pre-training. If training steps are held constant while context length increases, models with longer context see more total information despite equal gradient updates. Conversely, if training data is held constant, shorter-context models undergo more gradient steps. Since optimisation is non-linear, this comparison is also imperfect. In this section, we adopt a principled compromise where we pre-train models for the same number of steps while ensuring they observe identical numbers of unique tokens (and therefore equivalent total information). Specifically, we randomly sub-sample and expose the network to $42\%$ of the pre-training data for all token-matched experiments. The results in Figure 7 show that the same trends hold as in the main body of the paper. Increasing pre-training context continues to improve both downstream linear probing performance (with diminishing returns or regression at 150s), and zero-shot prediction of masked brain activity. Consistent results under this controlled setting generally rule out improvements due to additional information exposure.

## E. Tokenizer Comparison

As discussed in Section 2.1, we opted for the single-channel EEG pre-trained BioCodec tokenizer (Avramidis et al., 2025) over the MEG pre-trained BrainTokenizer (Xiao et al., 2025) because of BioCodec's ability to reconstruct our MEG data, even at 50Hz, with lower reconstruction error. This is likely due to the fact that BioCodec does not compress the channel dimension, entailing a trade off as it results in more tokens than BrainTokenizer. Nevertheless, as the field has not yet advanced to a stage where we know precisely which parts of the neural signal are relevant for speech decoding, opting for a tokenizer with lower reconstruction error helps avoid discarding task-relevant information from the signal. Although a tokenizer that compresses in time can learn cross-channel representations, we leave this task to our transformer backbone.

## F. Analysing Temporal Attention Heads

We analyse the temporal self-attention layers of our criss-cross transformer layers. For each temporal attention layer, we compute attention weights as $\mathbf{A} = \text{softmax}(\mathbf{Q}\mathbf{K}^\top/\sqrt{d})$, where $d$ is the head dimension.

We compute two complementary metrics to characterise attention patterns. The *mean attention distance* measures how far

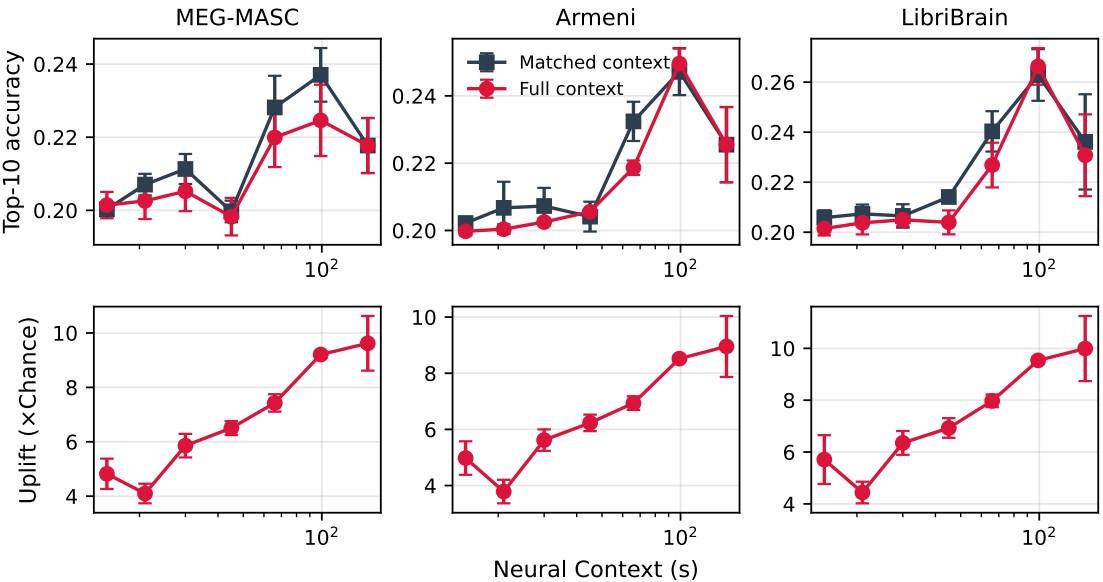

*Figure 7.* **(Top) Linear probing for word decoding with token-matched pre-training.** We pre-train models exactly the same way as described in the caption in Figure 4, except with fixed training tokens so that each pre-trained model sees exactly the same training data (token-matched). Here, we see similar trends with larger contexts systematically improving downstream performance. Diminishing returns, and potentially regression in performance, appear after 150s of pre-training context. The results are noisier as a consequence of three pre-training seeds (and linear probes) per context length instead of five due to computational constraints. **(Bottom) Zero-shot prediction of masked brain activity with token-matched pre-training.** The results here are collected the same way as described in the caption of Figure 5 (top), except with fixed training data for each pre-training run (token-matched). The trends remain very similar.

each query position attends on average:

$$\text{MAD}(t) = \sum_k A_{t,k} \cdot |t - k|, \tag{6}$$

where $A_{t,k}$ is the attention weight from query position $t$ to key position $k$. This metric is averaged across query positions and attention heads, then calculated for each layer. We report values in seconds, converting from timesteps by multiplying by $0.24 = r/f = 12/50$ where $r = 12$ is the tokenizer's downsampling ratio and $f = 50$ is the preprocessed sample rate of our data.

The *attention entropy* quantifies the uniformity of each attention distribution:

$$H(t) = -\sum_k A_{t,k} \log A_{t,k}. \tag{7}$$

Higher entropy indicates a more uniform attention distribution (attending broadly across positions), while lower entropy indicates concentrated attention on fewer positions. Both metrics are computed over 100 randomly sampled MEG segments from held-out data, with results aggregated across 5 random seeds per pre-training context length.

## G. Why Non-Invasive Decoding?

While intracranial approaches have been undeniably successful (Moses et al., 2021; Willett et al., 2023; Card et al., 2024), invasive brain-computer interfaces require craniotomy, carrying risks of infection, bleeding, and tissue damage. This is a significant concern for patients who are often already medically fragile due to conditions like ALS or locked-in syndrome. Implanted electrodes also degrade over time as scar tissue encapsulates the array, with signal quality declining over months to years and devices like Utah arrays eventually requiring replacement surgery. Beyond individual patient burden, invasive approaches face a scalability problem as one cannot easily, or ethically, implant thousands of subjects to build foundation models. Non-invasive methods like MEG and EEG allow large-scale data collection from healthy volunteers, enabling the pre-training/fine-tuning paradigm where pre-training data is, relatively speaking, abundant, and thus more easily scaled. They also lower the accessibility barrier allowing more volunteers to participate. The ethical case is also simpler as placing

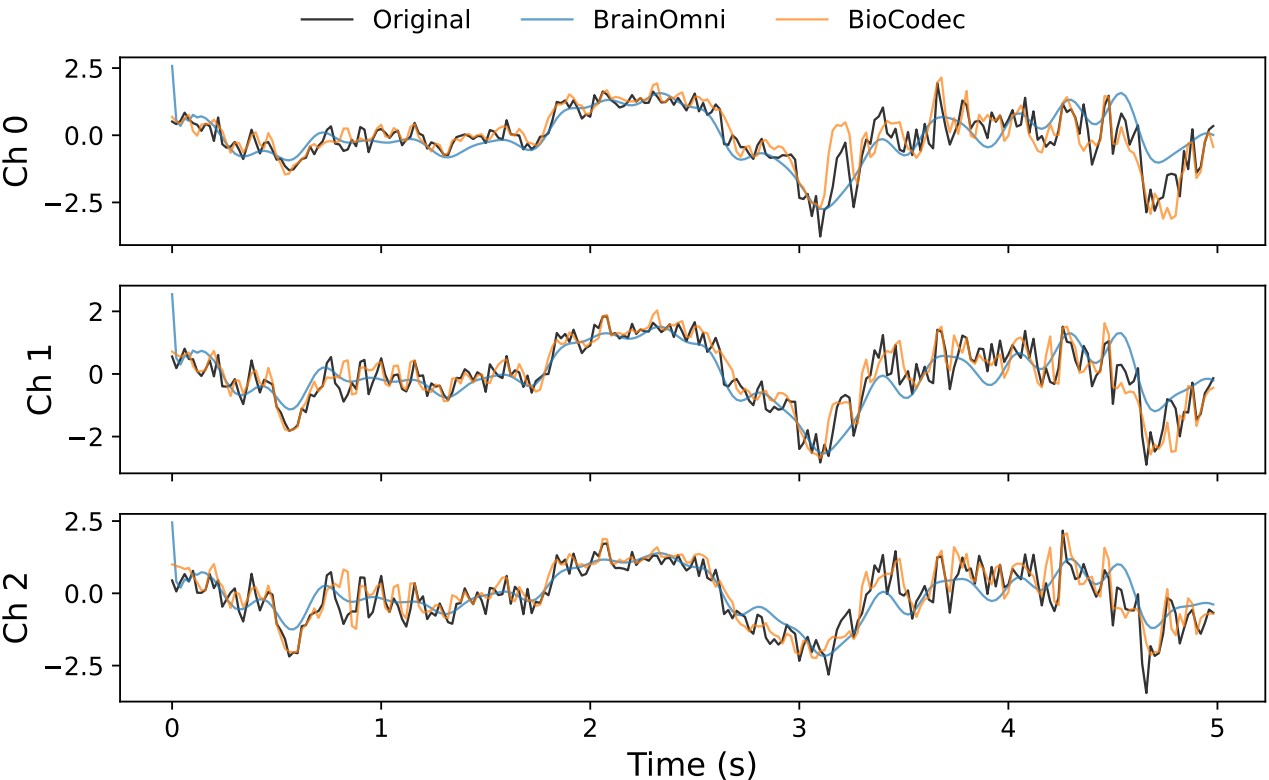

*Figure 8.* BioCodec vs BrainTokenizer (originating from BrainOmni). BioCodec reconstructs signals with lower reconstruction error (MSE of 0.41 vs 0.69). The plot shows a preprocessed 5-second sample taken from the MOUS dataset at 50Hz across three channels.

a helmet on someone's head is easier to justify than implanting a device in their brain. The obvious trade-off is the signal-to-noise ratio as intracranial recordings are orders of magnitude higher in fidelity. This has confined much prior non-invasive work to simpler speech sub-tasks such as speech detection (Dash et al., 2020; Jayalath et al., 2025b; Özdogan et al., 2025), phoneme recognition (Panachakel & Ramakrishnan, 2021; Gwilliams et al., 2022; Lee et al., 2025a; Moreira et al., 2025; Özdogan et al., 2025), keyword spotting (Elvers et al., 2025), semantic reconstruction (Tang et al., 2023), and isolated word classification (Défossez et al., 2022; Moreira et al., 2025). The challenge for non-invasive decoding is therefore to close the gap to invasive methods through better modelling, extracting more information from noisier signals, but with access to more data. Long-context pre-training, as explored in this work, represents a solution to this problem.

