# OpenReview forum: "MEG-XL: Data-Efficient Brain-to-Text via Long-Context Pre-Training"
_ICML.cc/2026/Conference — ICML 2026 regular_

### Official Review · Reviewer_sn1h · 2026-03-03

**Soundness:** 3
**Presentation:** 2
**Significance:** 3
**Originality:** 2
**Overall Recommendation:** 3
**Confidence:** 3

**Summary:**

MEG-XL introduces a long-context self-supervised pre-training framework for MEG-based brain-to-text decoding. While prior brain foundation models pre-train on 0.5–10 seconds of neural context, MEG-XL scales to 2.5 minutes per sample (5–300× longer), using a criss-cross transformer with factorized spatial-temporal attention and a frozen BioCodec RVQ tokenizer. The model is trained via masked token prediction with 3-second block masking on ~300 hours of MEG data from over 800 subjects.
#####
The paper contributes three findings. First, MEG-XL achieves strong data efficiency for word decoding: it significantly outperforms six brain foundation models in low-data regimes (47–57% top-10 accuracy with 13% training data versus near-chance for most baselines) and matches the supervised state-of-the-art using a fraction of subject data (1 hour vs 50 hours). Second, linear probing reveals that representations improve monotonically with pre-training context length, and models cannot leverage inference-time context beyond what they were pre-trained on — establishing long-context utilization as a learned capability. Third, interpretive analysis shows long-context pre-training induces hierarchical attention (local-to-global across layers) with lower entropy, indicating the model learns selective temporal attention patterns that short-context models never acquire.

**Compliance With Llm Reviewing Policy:**

Affirmed.

**Key Questions For Authors:**

Q1: The pre-training uses continuous 2.5-minute MEG, but fine-tuning uses concatenated 3-second word-locked epochs with temporal discontinuities at boundaries. How do you know the model is leveraging learned long-range neural dependencies rather than simply benefiting from having more context tokens for contrastive discrimination?
####
Q2: The 3-second masking block exactly matches the fine-tuning epoch duration. Have you tested other block sizes (1s, 5s, 10s)?
####
Q3: BIOT and EEGPT required channel reduction projections and shortened context (24s and full context respectively) due to architectural constraints, while LaBraM was limited to 15s. These modifications likely handicap their performance independent of pre-training quality. Could you report MEG-XL performance when similarly restricted to 15–24s context at fine-tuning time? If MEG-XL still outperforms under matched context constraints, the advantage is clearly representational; if not, the gains may be primarily architectural.
####
Q4: Performance drops substantially from 50-word to 250-word retrieval (e.g., MEG-XL on MEG-MASC: 47.0% → 16.8% at 13% data). What is the projected trajectory at clinically relevant vocabulary sizes (1k–10k words)? Even an extrapolation or preliminary experiment on larger vocabularies would help assess whether long-context pre-training provides a lasting advantage or whether the gains compress as task difficulty increases. This directly impacts the significance assessment.
####
Q5: Pre-training includes Dutch (MOUS) and Chinese (SMN4Lang) alongside English data, yet all evaluation is English. Have you ablated the contribution of non-English data?

**Limitations:**

The authors acknowledge key limitations including the focus on perceived rather than imagined speech, the 50-word vocabulary constraint, and that supervised methods win with abundant single-subject data. The Impact Statement appropriately raises privacy concerns around inferring mental content from brain activity and notes the gap to clinical deployment.
####
However, two areas could be strengthened. First, the paper does not discuss the practical accessibility barrier of MEG hardware itself — MEG systems cost millions of dollars, require magnetically shielded rooms, and are immobile, which limits the clinical populations who could benefit. Acknowledging this constraint and discussing potential transfer to more accessible modalities (e.g., EEG) would contextualize the practical significance more honestly. Second, the societal impact discussion could address the dual-use risk more concretely: as non-invasive decoding accuracy improves and data requirements shrink, the threshold for covert neural surveillance lowers, and the paper's explicit goal of reducing required training data accelerates this trajectory.

**Strengths And Weaknesses:**

Strengths:
####
Soundness: The experimental design is rigorous. The comparison against both supervised baselines (d'Ascoli et al., 2025) and six foundation models across three datasets spanning different data regimes (shallow multi-subject, deep single-subject) is thorough. The randomly-initialized MEG-XL control cleanly isolates pre-training gains from architectural contributions. The token-matched context scaling experiment (Appendix D) rules out the confound that longer-context models simply see more data. The linear probing analysis (Figure 4) with both matched and full context conditions convincingly establishes that long-context utilization is a learned capability, not an inference-time trick.
####
Presentation: The paper is clearly written with a well-structured narrative: motivation → method → comparisons → scaling analysis → interpretation. Figure 1 effectively communicates the core contribution at a glance. The three evaluation datasets are well-chosen to span clinically relevant axes (many subjects/shallow vs. single subject/deep). The paper is honest about limitations, acknowledging that supervised methods win with abundant single-subject data and that the vocabulary is limited to 50 words.
Significance: The clinical motivation is compelling — paralysed patients cannot provide extensive training recordings, making data efficiency essential. Reducing the data requirement from ~50 hours to ~1 hour represents meaningful practical progress. The finding that context utilization must be learned during pre-training (not just provided at inference) is a useful insight that could guide future brain foundation model design.
####
Originality: The core idea — scaling pre-training context from seconds to minutes for neural data — is a straightforward but well-motivated extension drawing a clear analogy from long-context language modelling. The combination of criss-cross attention, RVQ tokenization, and temporal block masking is a sensible engineering integration rather than a novel algorithmic contribution, but the resulting system substantially outperforms existing approaches.
####
Weaknesses:
####
Soundness: The 50-word retrieval vocabulary is small, and performance drops substantially with 250 words (Table 4), raising questions about scalability to practical vocabulary sizes. The fine-tuning input is not truly continuous MEG but concatenated 3-second word-locked epochs, creating a mismatch with the continuous pre-training regime that is acknowledged but not ablated. The pre-training data includes MOUS (Dutch) and SMN4Lang (Chinese), yet evaluation is English-only — whether multilingual pre-training helps or hurts is not disentangled. The BioCodec tokenizer was trained on EEG at 250Hz but applied to MEG at 50Hz; while reconstruction looks reasonable (Figure 8), the domain gap's impact on learned representations is not systematically evaluated. Statistical reporting uses standard error over only three seeds, which may understate variability.
####
Presentation: The paper could better discuss the concatenated-epoch input construction in the main text rather than briefly noting it is "not perfectly continuous." The relationship between pre-training masking block size (3s) and the word-locked epoch duration (3s) is convenient but the sensitivity to this alignment is not explored. Some baseline comparisons may be unfair — BIOT and EEGPT require channel reduction projections and shortened context due to architectural constraints, potentially handicapping their performance.
Significance: Absolute accuracy remains modest (47–63% top-10 over 50 words), far below clinical utility thresholds. The approach is demonstrated only for perceived speech, while the clinical target is imagined/attempted speech — a substantially harder problem where the benefit of long context is unclear. MEG hardware is expensive and immobile, limiting real-world deployment compared to EEG-based approaches.
####
Originality: The individual components (criss-cross attention, RVQ tokenization, masked prediction, contrastive fine-tuning) are all borrowed from prior work. The main novelty is simply using longer context windows, which is more of a scaling decision than a methodological contribution. The attention analysis (Section 3.4) provides descriptive characterization but no mechanistic explanation of what linguistic or neural structure the long context captures.

---

> ### Author Rebuttal · Authors · 2026-03-30
>
> Thank you for your detailed review. We ran new experiments to address your core concerns.
>
> **Baseline fairness (Q3).** To isolate representational quality from context, we fine-tuned MEG-XL restricted to 15s context on MEG-MASC at 13% data:
>
> | Model (Context) | Acc. |
> |---|---|
> | LaBraM (15s) | 33.2 |
> | BIOT (24s) | 20.0 |
> | MEG-XL (15s) | 36.9 |
> | MEG-XL (150s) | 47.0 |
>
> MEG-XL outperforms under matched context, confirming a representational advantage. The jump to 150s shows a strong context advantage too. Therefore, both factors contribute.
>
> **Concatenated epochs (Q1).** This follows d'Ascoli et al. (2025) for fair comparison. Non-word-locked decoding remains an open problem (see Sec. 2 in [Jayalath et al. 2025b]). The random-init control (Figure 3) addresses the mismatch concern as the gap between pre-trained and randomly initialised MEG-XL shows the gains come from learned priors, not the input format. Moreover, if the benefit were more tokens for contrastive discrimination rather than learned long-range priors, then giving a model pretrained with short (<150s) context more context (e.g. 150s) at fine-tuning would help. Figure 4 shows it does not. Therefore, the advantage comes from learning to leverage long-range neural dependencies.
>
> [Jayalath et al. 2025b]. Unlocking non-invasive brain-to-text. arXiv:2505.13446.
>
> **Masking block alignment (Q2).** We fine-tuned with 2.5s epochs (breaking the 3s alignment) on MEG-MASC at 13% data. Accuracy drops from 47.0% to 37.3%. This supports our view that alignment between pre-training and downstream structure is important (Section 3.3), but the misaligned model still outperforms all baselines. Part of the drop can also be attributed to 0.5s less information in each word-locked epoch.
>
> **Vocabulary scaling (Q4).** MEG-XL leads at both 50 and 250 words (Table 4). d’Ascoli et al. 2025 (Figure S5) show that the degradation should be log-linear as vocabularies expand and we expect MEG-XL to be similar. We note that invasive systems followed a similar path, achieving initial results on 50-word vocabularies before scaling to open-vocab over subsequent work (Moses et al. 2021; Willett et al. 2023). Moses et al. (2021) deployed a BCI with a 50-word vocab for an anarthria patient, showing it sufficient for a wide range of practical sentences (Supplement Section S5) and therefore clinically meaningful.
>
> **Multilingual pre-training (Q5), tokenizer domain gap, and seeds.** Multilingual pre-training has been shown to help (Jayalath et al. 2025; Xiao et al. 2025), so we treated it as a settled design choice. On the tokenizer, our results implicitly ablate this as BrainOmni uses a modality-matched tokenizer yet MEG-XL outperforms. In general, modality mixing has been shown to be helpful (Xiao et al. 2025). Tokenizer reconstruction fidelity (Appendix E) is a necessary condition for good pre-training and our results support this, though a full ablation would be ideal. We will add seeds in the camera-ready though note that effect sizes are large with non-overlapping error bars and Welch's t-tests reach p < 0.05 consistently.
>
> **Hardware and clinical gap.** We fine-tuned MEG-XL on Broderick et al.'s (2018) EEG dataset despite MEG-only pre-training, achieving 38.9% (vs. 24.4% for training a random-init model). This demonstrates cross-modality transfer to cheaper hardware. We acknowledge the gap to clinical deployment (Section 4) in terms of perceived vs imagined speech and clinical accuracy, though this gap is shared by all current non-invasive decoding work. Our contribution is narrowing the data-efficiency bottleneck rather than claiming immediate clinical readiness.
>
> **Originality.** The contribution is empirical and scientific instead of algorithmic. We observe that underlying long-range dependencies are unexploited in neural data. We demonstrate long-context utilisation is learned and models cannot exploit context beyond pre-training length (Figure 4). Our results show long-context pre-training then improves results substantially. This mirrors findings in LLMs (Transformer-XL/LongFormer) but in a new domain and fundamentally changes how brain foundation models should be trained.
>
> **Linguistic feature interpretability.** Mechanistic analyses of language encodings are important future work. d’Ascoli et al. 2025 (Figure 6C and 6D) identify the linguistic features encoded during contextual word decoding. As MEG-XL’s fine-tuning objective is identical, it should indicatively lead to similar encodings.
>
> We will make the concatenated-epoch construction more prominent in the main text, including clarification of continuous-vs-concatenated in the abstract, and expand the dual-use discussion in the Impact Statement. We believe the new experiments, context-restricted comparison, masking alignment ablation, and cross-modality transfer, should address your core concerns.
>
> We hope that our responses and new experiments have answered your concerns. Is there anything else we can address?

---

> > ### Author Rebuttal · Reviewer_sn1h · 2026-04-03
> >
> > Thank you for your feedback.

---

> > > ### Author Response · Authors · 2026-04-03
> > >
> > > Thank you for acknowledging that your concerns have been fully resolved. We appreciate the time you invested in reviewing our work.
> > >
> > > We noticed that your score currently remains at 3 (Weak Reject). Given your acknowledgement that concerns have been fully resolved, would you consider updating your score to reflect this?

---

### Official Review · Reviewer_KVwF · 2026-03-12

**Soundness:** 3
**Presentation:** 3
**Significance:** 3
**Originality:** 3
**Overall Recommendation:** 4
**Confidence:** 3

**Summary:**

This paper introduces MEG-XL, a self-supervised framework for brain-to-text decoding using magnetoencephalography (MEG). The paper focuses on long-context pre-training to help model long-range dependencies in brain activity that short-window approaches cannot. Specifically, the paper introduces MEG-XL, which models 2.5 minutes of neural context per sample (approximately 191k tokens) to enable the model to capture long-range dependencies in neural responses to speech. The model is pre-trained using masked token prediction on large multi-subject MEG datasets and later fine-tuned for word decoding tasks. Experiments across three datasets (MEG-MASC, Armeni, LibriBrain) show that long-context pre-training significantly improves performance when limited fine-tuning data is available.

**Compliance With Llm Reviewing Policy:**

Affirmed.

**Key Questions For Authors:**

1. Is the improvement in performance mainly due to context length or pre-training dataset scale?

2. How sensitive is performance to the tokenizer? Since the tokenizer is trained on EEG data rather than MEG, how robust are the results to different tokenization strategies?

3. Does long-context modeling generalize across neural recording modalities?

4. Can the model scale to open-vocabulary decoding? How does performance change when increasing or decreasing vocabulary sizes?

**Limitations:**

1. The current work is primarily evaluated on word decoding, and it is unclear whether the approach or the learned representations generalize to other neural tasks.

2. There is a potential for overfitting to structured speech stimuli in the paper, as the datasets use structured speech stimuli (e.g., audiobooks), which may not reflect natural conversational speech.

**Strengths And Weaknesses:**

Strengths

1. The paper aims to address a major challenge in brain-computer interfaces, which is increasing training data efficiency for brain-to-text decoding. This is particularly important for clinical populations that cannot provide extensive training data, and is an impactful research problem and application.

2.  The core idea of scaling neural context to minutes rather than short windows in seconds is an interesting contribution. This also aligns with recent research trends in long-context modeling in language and audio.

3. The model is evaluated across three distinct MEG speech datasets in different splitting regimes, which include shallow multi-subject, deep single-subject, and somewhere in between.

4. The analysis of hierarchical attention patterns emerging with long-context pre-training provides useful insight into what the model learns.

Weaknesses

1. The method is tailored specifically to MEG recordings, which are expensive and less widely available than EEG. I am wondering if the author has considered experimenting with EEG-friendly tasks.

2. There is a limited vocabulary in evaluation. The decoding task is restricted to a 50-word vocabulary, which is far from realistic communication settings. But will the model behave differently by increasing or decreasing this choice?

3. There is limited interpretability of neural representations, even though attention analyses are provided, the paper does not explain what linguistic or neural features are actually encoded by the model.

4. It is an interesting idea that the author uses BioCodec, trained on EEG signals, to tokenize MEG signals, but this may still introduce potential modality mismatch.

---

> ### Author Rebuttal · Authors · 2026-03-30
>
> Thank you for your detailed review and positive feedback. Below, we have responded to your concerns:
>
> **Regarding weakness 1**: We ran a new fine-tuning experiment on Broderick’s 2018 EEG speech dataset and find that MEG-XL, despite its MEG pre-training, generalises well, achieving 38.9% top-10 balanced accuracy with a 50-word retrieval set (where training a random-init model gets 24.4%). We hope this helps to alleviate the reviewer’s concern about EEG generalisation.
>
> Generally, we chose to focus on MEG as this is the modality that has afforded recent breakthroughs in non-invasive electrophysiological speech decoding, e.g. d’Ascoli et al. 2025 and Défossez et al. 2023. While SQUID-MEG is more expensive and less widely available, new OPM-MEG scanners are cheaper and more portable.
>
> **Regarding weakness 2**: With a larger 250-word vocab, MEG-XL still outperforms all other baselines (see Table 4 in our Appendix). Performance generally degrades log-linearly with larger vocabs and is a known limitation (see d’Ascoli et al. 2025 Figure S5). Our primary concern is locked-in patients who otherwise cannot communicate. For these patients, a 50-word vocabulary can provide meaningful communication. This was the basis of Moses et al. 2021 in the invasive domain, who used a 50-word vocabulary with an anarthria patient. In their supplement, Section S5 shows that a 50-word vocab is sufficient to construct a wide range of useful sentences.
>
> **Regarding weakness 3**: Linguistic feature analysis is important future work. Our attention analysis (Section 3.4) provides complementary architectural interpretability. d’Ascoli et al. 2025 (Figure 6C and 6D) identify the linguistic features encoded during contextual word decoding. As MEG-XL’s fine-tuning objective is identical, it should lead to similar encodings.
>
> **Regarding weakness 4**: While MEG-XL still outperforms all alternatives, we think that fine-tuning the tokeniser with MEG may improve results further. Recent evidence suggests the modality gap may be small, however. Our new EEG transfer results (first paragraph) show strong transfer across modalities and Xiao et al. 2025 find MEG/EEG representations can be mixed to improve performance on both modalities.
>
> **In response to question 1**: The improvement in performance is largely due to context length rather than pre-training scale. Crucially, we conduct a token-matched ablation in Appendix D which directly controls for data scale and still shows context length is what improves results. Additionally, the closest comparable work to ours is BrainOmni, which pre-trains with a much larger volume of data. Our pre-training datasets are a subset of theirs, representing less than half of their MEG pre-training data in terms of recording hours.
>
> **In response to question 2**: Our results implicitly ablate tokenizer choice as we compare against BrainOmni, who use a tokenizer trained with MEG. While properly ablating this with pre-training runs is prohibitively expensive, an (imperfect) proxy for tokenizer effect is tokenizer reconstruction loss. In Appendix E, we show the BioCodec tokenizer preserves more of the original signal in reconstruction than the BrainTokenizer. We agree that a proper ablation would be ideal, however.
>
> **In response to question 3**: **Yes.** Please see above for our new experiment fine-tuning MEG-XL on EEG data.
>
> **In response to question 4**: Please see our response to weakness 2.
>
> **Regarding limitation 1**: Evidence in our paper suggests the representations do generalise. Pre-training has no exposure to word labels and is mostly from motor tasks. That these representations then transfer well to word decoding is evidence of generality. This is also shown through our zero-shot masked token prediction results (Figure 5, top). If the model were capturing only speech-specific patterns, we would not expect systematic gains on this generic objective over held-out data. We fully agree that explicit evaluation on other downstream tasks would strengthen the claim and plan to pursue this in future work.
>
> **Regarding limitation 2**: We appreciate this concern. MEG-MASC, Armeni, and LibriBrain are the three largest English-language perceived speech MEG datasets available and represent the current evaluation standard for non-invasive speech decoding (see d'Ascoli et al., 2025 and Défossez et al. 2023). This constraint is not unique to our method but to the subfield. Evaluation on conversational MEG would be valuable and hope the community produces such datasets. We are also actively working on collecting such a dataset ourselves.
>
> We hope that our responses and new experiments have resolved your concerns. Is there anything else we can address?

---

> > ### Author Rebuttal · Reviewer_KVwF · 2026-04-02
> >
> > The author has answered most questions, and I will keep my score.

---

> > > ### Author Response · Authors · 2026-04-03
> > >
> > > Thank you for confirming our concerns are resolved. We would like to share one final experiment we started after the first round of responses that addresses **limitation 2 (structured stimuli)**.
> > >
> > > **New experiment with natural conversational speech.** We evaluated MEG-XL on the English subset of Kymata Soto [A], a recently released (Jan 2026) 20-subject conversational MEG dataset. This is natural dialogue rather than audiobooks and was designed for analysis of conversational speech. Critically, each subject has only seven minutes of stimulus data from multi-participant conversations, far less than any dataset in our paper. MEG-XL achieves 44.5% top-10 accuracy (vs 25.6% when training from random initialisation), demonstrating that (a) the approach generalises beyond structured speech to conversation, and (b) data efficiency holds even with less data per subject than MEG-MASC. We believe this resolves what you identified as a remaining weakness limiting the paper's impact.
> > >
> > > Given that you noted all concerns are fully resolved, would you consider raising your score?
> > >
> > > [A] Yang, C., Parish, O., Klimovich-Gray, A., Wingfield, C., Marslen-Wilson, W.D., Zhang, C., Woolgar, A. and Thwaites, A., 2026. Kymata Soto Language Dataset: an electro-magnetoencephalographic dataset for natural speech processing. Scientific Data.

---

### Official Review · Reviewer_vcDz · 2026-03-12

**Soundness:** 3
**Presentation:** 4
**Significance:** 3
**Originality:** 3
**Overall Recommendation:** 5
**Confidence:** 5

**Summary:**

This paper introduces MEG-XL, which is a model pre-trained on much longer context (2.5 minutes) than previous MEG models in order to learn better representations for word decoding. The paper finds fine-tuning MEG-XL word decoding accuracy that is comparable to both supervised models and foundation models, and with much less data. The paper also seeks to understand how and why this long-context pretraining improves downstream task performance.

**Compliance With Llm Reviewing Policy:**

Affirmed.

**Final Justification:**

MEG-XL is an important contribution to the field, in identifying a better way to do pre-training for MEG data (with demonstrated and detailed experiments on multiple downstream tasks). The authors addressed all my concerns in the rebuttal, and I'm happy to keep my score as an Accept.

**Key Questions For Authors:**

- Could the gap in the tokenizer pretraining sampling frequency, and the downsampled frequency here be responsible for some of the performance differences seen between MEG-XL and d’Ascoli et. al? Did the authors consider fine-tuning the tokenizer to match the lower sampling rate?
- Would PEFT help MEG-XL overcome the the gap on “deep” datasets?
- Did the authors consider other more efficient architectures like Mamba2?

**Limitations:**

Yes

**Strengths And Weaknesses:**

Strengths:
- This paper is technically very sound, and proposes a principled novel approach to long-context MEG modeling using established techniques in other (M/E)EG foundation model papers (CBraMod, BioCodec, etc.). The paper also evaluates the models on multiple evaluation tasks, both on full fine tuning and linear probing. The authors also try to understand what their approach actually accomplishes and why.
- MEG-XL achieves competitive performance with less subject data across 3 different downstream tasks, and achieves SOTA performance compared to pretrained foundation models, even on fully end-to-end fine-tuning. The authors also provide a clear explanation for why the performance on each downstream dataset is the way that it is.
- The paper demonstrates clearly that more context improves performance on word decoding, and on zero-shot prediction. Furthermore, the authors highlight how the model understands the long context (early layers are local, and later layers are global)
- The paper is well written and clear in its descriptions of the methods, data, and results.
- The paper provides several useful ablations (like resampling to 100Hz, and token matched pretraining) that strengthen the story.

Weaknesses:
- The word-locked decoding approach concatenates non-continuous segments of MEG data, potentially limiting the model from learning or utilizing true long range features in the signal.
- As the authors acknowledged, downsampling to 50 Hz may be limiting the model from utilizing speech related signals. This downsampling may be especially exacerbated when considering that BioCodec was trained on 250 Hz EEG data.
- Performance improvements plateau after 100s of context, perhaps the other 50s of context adds more computational overhead than real value.
- Model performance suffers on certain downstream tasks with “deep” data on a single subjects.

---

> ### Author Rebuttal · Authors · 2026-03-30
>
> Thank you for your thorough and positive review. We have addressed your questions below.
>
> We would first like to highlight a result we believe strengthens the significance of this work. **MEG-XL transfers from MEG to EEG**, achieving 38.9% on the Broderick et al. 2018 EEG dataset (where training a random-init model gets 24.4%), despite exclusively MEG pre-training. This cross-modality transfer suggests long-context pre-training may serve as a general foundation for non-invasive neural decoding not just MEG-specific word decoding. The result supports prior work showing MEG/EEG transfer in simpler tasks (Xiao et al. 2025). This is promising since clinical application of EEG is cheaper and simpler than MEG.
>
> **Re: tokenizer / frequency gap.** The d'Ascoli et al. (2025) baseline uses the same preprocessing, where sampling frequency is identical, and cannot explain performance differences between them. However, as you point out, the tokenizer mismatch might explain it, though we suspect it may be more to do with capacity differences (see next answer below). We agree that fine-tuning BioCodec at 50Hz would likely help. Our Appendix E results suggest the current reconstruction quality is already strong but this may further improve performance and we will try this experiment next.
>
> **Would PEFT help on deep datasets?** With abundant single-subject data, the random-init model catches up to the pre-trained model (Figure 3). Therefore, the gap is unlikely to be about how we adapt (which PEFT addresses) but about model capacity. The gap to d'Ascoli et al. (2025) may be architectural as their 200M-parameter model has 10x our backbone’s capacity. MEG-XL prioritises transferability over deep data specialisation, where ours is the clinically relevant regime. For a universally state-of-the-art model, we would need to scale up MEG-XL parameters to match and are working to acquire greater compute resources to achieve this.
>
> **Mamba2 / efficient architectures.** Criss-cross attention was chosen because it factorises naturally along spatial and temporal axes, which is architecturally meaningful for MEG. Mamba2's linear-time scaling does not have this property and therefore was not our first choice, but would be attractive for pushing beyond 150s. As brain-to-text performance seems to plateau after 100s, we did not feel the need to push further. However, for tasks where further context may help, we expect our framework to be architecture-agnostic (within computational limits) and Mamba2 is a promising alternative architecture to try.
>
> On the weaknesses raised:
>
> **Re: concatenated epochs.** Our random-init control (Figure 3) shows the gains are from pre-trained priors rather than the input format. Our other control (Figure 4) shows that models pre-trained with shorter contexts do not benefit from longer contexts downstream. Together, this shows that downstream gains emerge from pre-training which helps to leverage long-range features. Therefore, the model is utilising true long-range features in the signal despite the non-continuous segments of MEG data.
>
> **Re: downsampling to 50Hz.** Please see our second paragraph above.
>
> **Re: the 100s plateau.** This applies to word decoding specifically but not all tasks. In zero-shot masked prediction, MEG-XL continues improving through 150s (Figure 5), suggesting additional context carries structure that other tasks may exploit by fine-tuning MEG-XL. Therefore, the additional computational load is worth it when pre-training MEG-XL as other tasks could still exploit it. For word decoding, practitioners can choose to fine-tune with 100s of context (rather than 150s) if they wish to reduce computational overhead.
>
> **Re: deep single-subject performance.** This is the expected regime where pre-training helps least, as abundant subject data substitutes for learned priors. We see the clinical regime (scarce data, many subjects) as the high-impact target, where MEG-XL's advantage is largest. Scaling MEG-XL's parameters (as discussed in the PEFT answer) to match d’Ascoli et al. 2025 would likely close this gap however.
>
> We hope that our responses and new experiments have resolved any remaining concerns. Is there anything else we can address?

---

> > ### Author Rebuttal · Reviewer_vcDz · 2026-04-02
> >
> > Thank you for acknowledging my concerns! I'm very happy to recommend this paper be accepted to ICML, and will keep my score.

---

### Decision · Program_Chairs · 2026-04-30

**Decision:**

Accept (regular)

**Comment:**

The manuscript demonstrates that long context pretraining when decoding text from neural recordings leads to improved performance.

Reviewers found that the paper is well-motivated, that the contribution is interesting, the evaluation is well done across many datasets and tasks, and reasonable baselines are considered. Prior work has not trained models with long context of this form and as a consequence are likely overlooking important dynamics in neural recordings.

In the text universe long context has expanded to mean hundreds of thousands to millions of tokens, the method here plateaus very early. This may be because of the approach taken which stitches together pieces of individual words to create the longer contexts. The segmented approach is the biggest weakness of this work. This likely means that long-range neural dynamics, i.e., the temporal evolution of the system, are muddled.

Brining long context to this space is something that others can build on and that will be of interest to the community regardless of any limitations.